# Information Diffusion Prediction with Graph Neural Ordinary Differential Equation Network

## ABSTRACT

Information diffusion prediction aims to forecast the path of information spreading in social networks. Prior works generally consider the diffusion process to be driven by user correlations or preferences. Recent works focus on characterizing the dynamicity of user preferences and propose to capture users' dynamic preferences by discretizing the diffusion process into structure snapshots. Despite their effectiveness, these works summarize user preferences from partially observed structure snapshots, ignoring that users' preferences are evolving constantly. Moreover, discretizing the diffusion process makes these models overlook abundant structure information across different periods, reducing their ability to discover potential participants. To address the above issues, we propose a novel **G**raph Neural **O**rdinary **D**ifferential **E**quation **N**etwork (GODEN) for information diffusion prediction, which incorporates neural ordinary differential equations (ODE) to model the continuous dynamics of the diffusion process. Specifically, we design two coupled ODE functions on nodes and edges to describe their co-evolution dynamic and infer user dynamic preferences based on the solution of ODEs. Besides, we extract user correlations from a heterogeneous graph to complement user encoding for prediction. Finally, to predict the future user infections of the observed cascade, we represent its diffusion pattern in terms of user and temporal contexts and apply a multi-head attention module to attend to different contexts. Experimental results confirm our approach's effectiveness on four real-world datasets, with our model outperforming the state-of-the-art diffusion prediction models.

## CCS CONCEPTS

• **Computing methodologies → Neural networks**; • **Information systems → Social networks**.

## KEYWORDS

social network, information diffusion prediction, graph neural network, ordinary differential equations

**ACM Reference Format:**
. 2024. Information Diffusion Prediction with Graph Neural Ordinary Differential Equation Network. In *Proceedings of ACM Conference (Conference'17)*. ACM, New York, NY, USA, 10 pages. https://doi.org/10.1145/nnnnnnn.nnnnnnn

**Unpublished working draft. Not for distribution.**

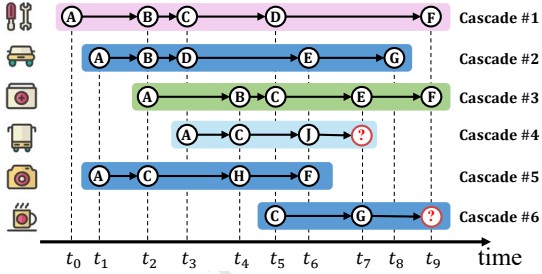

**Information diffusion cascades**

**Figure 1: A simple example of information diffusion cascades. Icons on the left represent the topic of cascades. Suppose the overall diffusion process is split into three diffusion periods** $[t_0, t_4)$**,** $[t_4, t_7)$**,** $[t_7, t_9)$**.**

## 1 INTRODUCTION

In recent years, multimedia social networks have become indispensable ways to publish and distribute information. Massive users interact with others online, facilitating the rapid dissemination of information and forming information cascades [37]. Effective prediction of future participants in information cascades has become a challenging but critical task for numerous social applications, such as social recommendation [13, 30] and disinformation control [29].

Information diffusion prediction problems have spurred significant research interest for decades. Existing works can be principally summarized into three categories. 1) *Feature engineering-based models* [1, 34] assume that the diffusion process abides by predefined diffusion functions or models. They generally extract representative features of users and cascades to compute the diffusion probabilities and fit them with predefined diffusion functions or models at the macro level. However, these models are hardly generalized to different domains due to their restrictive assumptions. 2) *Sequence-based models* [10, 26, 32] model cascades as sequence data and exploit sequence models, *e.g.*, RNN or attention layer, to extract user correlations within diffusion paths. Despite their progress, their emphasis on sequential data caused them to neglect the impact of social relations, failing to extend user correlations beyond sequences. 3) *Graph-based models* [22, 27] introduce various graph structures to extend user correlation, such as social networks and diffusion networks. Most recently, some researchers [19, 31, 35] find that user preferences also have a crucial impact on the diffusion process. Considering the dynamic nature of users' preferences, they construct structure snapshots to discretize the diffusion process into multiple periods and introduce graph neural networks(GNNs) to describe users' preferences at each period, which achieves encouraging prediction performance.

However, current models with dynamic graphs focus on summarizing users' preferences at each period based on partially observed structure snapshots, which brings two natural deficiencies. For one,

current models ignore the fact that users' preferences are changing constantly. Users may participate in different cascades in the same diffusion period and show their varying preferences, *e.g.*, user A at the period $[t0, t4]$ in Figure 1. Summarizing all user interactions at certain diffusion periods to describe their preferences brings more harm than good in prediction. For example, when we predict the subsequent users in Cascade #4, fusing users' preferences can hardly achieve prediction, since only users A participates in relevant cascade in period $[t0, t4]$. For another, current models concentrate on partially observed structures in each period, making them neglect abundant user correlation across different periods. As shown in Figure 1, when predicting the future diffusion trends in Cascade #6 after timestamp $t_7$, if we only consider user interactions within each diffusion period, we find User $G$ interacts with nobody. However, from a global perspective, we could observe diffusion structure $C \rightarrow D \rightarrow F$, $C \rightarrow E \rightarrow F$, and $C \rightarrow H \rightarrow F$ from Cascade #1, Cascade #3, and Cascade #5, indicating that user $F$ is a potential participant.

To solve the above problems, we propose a novel **G**raph neural **O**rdinary **D**ifferential **E**quation **N**etwork (short for **GODEN**) for information diffusion prediction. The key idea of GODEN is to characterize the evolutionary dynamics of the diffusion process based on graph structures. To achieve this goal, we first leverage three types of relations to comprehensively characterize the diffusion process and apply GNN to obtain the initial state of users. Then, we design two coupled ODEs on users and their relations to characterize their co-evolution dynamics since they are deeply correlated in the diffusion process. We apply a channel attention mechanism to infer users' dynamic preferences based on the solution of the ODE function. Besides, we apply a graph neural network to capture users' static correlations from a global perspective to complement the users' dynamic preferences. To predict future user infections, we first represent the diffusion pattern of the observed cascade based on its user context and temporal context. Then, we apply a multi-head self-attention mechanism to attend to different contexts and solve the information diffusion prediction problem.

Our main contributions are summarized as follows:

- We propose two coupled ODEs to learn the evolution pattern for users and relations in the diffusion process and infer users' dynamic preferences.
- We extract users' static correlations to extend their dynamic preferences and represent the specific diffusion pattern of cascades by learning the user context and temporal context information to promote the prediction.
- We conducted extensive experiments on four public datasets. The results show that GODEN outperforms state-of-the-art models on the information diffusion prediction task, demonstrating its effectiveness.

## 2 RELATED WORK

### 2.1 Information Diffusion Prediction

The information diffusion prediction task aims to predict future user infections based on historical diffusion paths, which have been widely studied over decades. Existing diffusion prediction models are mainly categorized into three categories: (1) feature engineering models, (2) sequence-based models, and (3) graph-based models.

Early feature-engineering models believed the diffusion process adheres to specific diffusion models, such as the independent cascade model [11] and epidemic models. However, their stringent assumptions constrain their ability to characterize complex diffusion patterns in the real world. Some embedding-based models further improve these models. They encode users as embeddings by maximizing specific diffusion functions and infer future diffusion probability through vector calculations. Although effective, this class of methods still hardly generalizes to different social scenarios.

Sequential-based models emerged along with the fast development of deep learning. They [10, 23, 32] treat diffusion cascades as sequences and regard the information diffusion prediction as a sequential prediction task. Various sequential models, *e.g.*, LSTM [8] or attention mechanisms [21], are applied to capture sequential and temporal features in information diffusion. NDM [32] employs multi-head attention mechanisms to model the diffusion sequence and incorporated convolution neural networks (CNNs) to alleviate long-term memory loss in sequential modelling. HiDAN [28] builds a hierarchical attention network to jointly capture user dependency and the time decay effect in the diffusion sequence. However, users' social structures, as one of the critical channels of information diffusion, are generally overlooked by sequential-based models.

Due to the recent success of graph neural networks (GNNs), graph-based models have demonstrated their effectiveness in tasks of diffusion prediction. Various graph structures, mostly social graphs, are also widely exploited to extract non-sequential user correlations. SNIDSA [27] builds a novel recurrent model to jointly incorporate social information from users and sequential information from cascades. FOREST [33] encodes historical information in cascades with GRU and combines it with neighbour information from the social structure. CEGCN [22] jointly models users and cascades in the same heterogeneous graph and extracts collaborative diffusion patterns via graph neural networks. Most recently, some studies [24, 35] have found that users' preferences play a critical role in facilitating information diffusion. Since users' preferences change as time passes, they model the diffusion process as a series of structure snapshots and employ graph neural networks to capture users' dynamic preferences. DyHGCN [35] extracts neighbour influence and diffusion preferences as users' dynamic preferences via a heterogeneous GCN. MS-HGAT [19] introduces a sequential hypergraph to capture users' interaction preferences and integrate them with static social relations to predict information diffusion.

Despite effectiveness, models with dynamic graphs simply summarize users' preferences based on observed structure snapshots from each diffusion period, which ignores the continuous evolution of users' preferences and the abundant structure information across different periods.

### 2.2 Graph Neural Networks

GNN is a class of neural networks that operate directly on graph-structured data and have shown remarkable performance in various domains. Recently, some researchers have extended GNNs to dynamic domains to capture the chronological characteristics of graph structures. These models are generally split into discrete-time dynamic GNNs (DTDGNNs) and continuous-time dynamic GNNs

(CTDGNNs). DTDGNNs [4, 17] discretize dynamic graphs as multiple structure snapshots and apply static GNN to process each snapshot for node representation. For CTDGNNs [20, 25], they represent dynamic graphs as a series of node interactions with precise timestamps in chronological order. They design specific recurrent modules to aggregate historical messages to update the node state.

For diffusion prediction, existing dynamic-graph-based works rely on DTDGNNs to summarize the users' preferences at different periods, failing to consider the continuous evolution of user preferences. Unlike existing works, we consider modelling the continuous evolution of users' preferences with ODEs.

## 2.3 Ordinary Differential Equation

Neural ODEs [3] have been proposed as a new paradigm for generalizing discrete deep neural networks to continuous-time scenarios. They specify the dynamics of the hidden state using a neural network $f_\theta$ with parameters $\theta$. Given an initial state $\mathbf{x}(0)$ They define a hidden state $\mathbf{x}(t)$ as a solution to the ODE initial-value problem (IVP):

$$\mathbf{x}(t) = \mathbf{x}(0) + \int_0^t \frac{d\mathbf{x}}{d\tau} d\tau = \mathbf{x}(0) + \int_0^t f_\theta(\mathbf{x}(\tau), \tau) d\tau \quad (1)$$

$\mathbf{x}(t)$ can be evaluated at any desired time through a numerical ODE solver without any internal operations [3], which allows Neural ODE to be built as a block for the whole neural network. Due to their superior performance and flexible capability, neural ODEs have been widely adopted in various research fields, such as traffic flow forecasting [6] and sequential recommendation [16]. Recently, some advanced methods connect GNNs with neural ODEs. GODE [14] generalizes the concept of continuous-depth models to graphs and parameterizes the derivative of hidden node states with GNNs. Inspired by the outstanding performance of ODEs in dynamic systems, we introduce ODEs to model the continuous dynamics in the diffusion process.

## 3 PRELIMINARY

### 3.1 Problem Formulation

Normally, an information diffusion process is recorded as a cascade $c_m = \{(u_1^m, t_1^m), (u_2^m, t_2^m), ..., (u_{L_m}^m, t_{L_m}^m)\}$ in chronological order, where element $(u_i^m, t_i^m)$ denotes that user $u_i^m$ performs an action to participation $c_m$ at time $t_i^m$, e.g., forwarding a Twitter message. $L_m$ is the maximum cascade length. The cascade $c_m$ can be further divided into user sequence $c_m^u = \{u_1^m, u_2^m, ..., u_{L_m}^m\}$ or timestamp sequence $c_m^t = \{t_1^m, t_2^m, ..., t_{L_m}^m\}$, if we focus only on the orders of users or the temporal information in the cascade $c_m$. We collect all historical cascades and users in $C = \{c_1, c_2, ..., c_{|C|}\}$ and $\mathcal{U} = \{u_1, u_2, ..., u_{|\mathcal{U}|}\}$, respectively. Moreover, describing and quantifying various user relations in the diffusion process is essential to the information diffusion prediction task. We then introduce graph structures used in the paper: social graph $\mathcal{G}_s$, diffusion graph $\mathcal{G}_d$, and bipartite graph $\mathcal{G}_b$. They are shown in the right of Figure 2. The social graph $\mathcal{G}_s = \{\mathcal{V}_s, \mathcal{E}_s\}$ is a directed graph that describes The social connections among users. $\mathcal{V}_s$ is the set of nodes representing social users. $\mathcal{E}_s$ is the set representing users' social relations. If the following relation exists from user $u_i$ to user $u_j$, a directed edge $u_i \rightarrow u_j$ will be added to edge set $\mathcal{E}_s$. Similarly, the diffusion

graph $\mathcal{G}_d = \{\mathcal{V}_d, \mathcal{E}_d\}$ is a directed graph formed by users' diffusion connections. $\mathcal{V}_d$ is the node set representing users in the historical cascades. $\mathcal{E}_d$ is the edge set representing diffusion actions. If we observe user $u_i$ forward information from user $u_j$, there is a directed edge $u_i \rightarrow u_j$ in the diffusion graph. The bipartite graph describes the connection between cascades and their corresponding users. $\mathcal{V}_d$ is the node set that contains both user and cascade nodes. $\mathcal{E}_d$ is the edge set representing the connection between users and cascades. we add a directed edge between each cascade node and its users, i.e., $u_i \rightarrow c_i, u_i \in c_i$.

Based on the above introductions, we describe the task of **information diffusion prediction** as: given the set of user $\mathcal{U}$, the set of historical cascades $C$, and an observed cascade $c_o = \{(u_i^o, t_i^o)|u_i^o \in \mathcal{U}, i < L_{c_o}\}$. $L_{c_o}$ refers to the maximum length of $c_o$. Our goal is to compute the conditional probability $\hat{y}_j = p(u_j|c_o)$ to show how likely user $u_j$ will participate in this cascade at the next timestamp.

## 4 METHOD

This section introduces our graph neural ordinary differential equation network (GODEN). The overall architecture of GODEN is shown in Figure 2, which has three major components: 1) User encoding module, which generates user embeddings by modeling the dynamic evolution of the diffusion process and capturing users' static correlations. 2) Cascade representation module, which represents the diffusion pattern based on the temporal and user contexts in the observed cascade. 3) Prediction module, which applies a multi-head attention module to calculate the infection probability of candidates.

### 4.1 User Encoding

*4.1.1 Dynamic Preference Encoding.* The information diffusion process is affected by different factors. To comprehensively describe the diffusion process, we first merge social graph $\mathcal{G}_s$, diffusion graph $\mathcal{G}_d$, and bipartite graph $\mathcal{G}_b$ into heterogeneous graph $\mathcal{G}_h = \{\mathcal{V}_h, \mathcal{E}_h, \mathcal{W}_h\}$. $\mathcal{V}_h = \{v_h|v_h \in \mathcal{V}_u \cup \mathcal{V}_b\}$ is the set of nodes constructed by both the user nodes and the cascade nodes. $\mathcal{E}_h = \{\mathcal{E}_s \cup \mathcal{E}_d \cup \mathcal{E}_b\}$ is the edge set with three types of relations. $\mathcal{W}_h = \{e_h|e_h \in \mathcal{E}_h\}$ is the set of edge weights to differentiate edge effects. We initialize the weight of all edges as 1.

To infer the users' dynamic preferences, we first encode both users and cascades into embeddings as their initial state in the diffusion process. We apply a GNN layer for user encoding and design different aggregation strategies for user and cascade nodes since they share different neighbor contexts in heterogeneous graph $\mathcal{G}_h$.

Specifically, for user node $u_i$ at the $(l+1)^{\text{th}}$ GNN layer, its neighbor contexts contain both user and cascade nodes. Therefore, we divide the neighbors by the edge types and separately aggregate

**Figure 2: The overview architecture of GODEN.**

their contextual information, which is formulated as,

$$
\mathbf{a}_{\mathcal{N}_u}^{u_i(l+1)} = f\left(\frac{e_h^{ij}}{|\mathcal{N}_u|} \sum_{u_j \in \mathcal{N}_u} \mathbf{W}_{u_j}^{u(l+1)} \mathbf{x}^{u_j(l)}\right),
$$

$$
\mathbf{a}_{\mathcal{N}_c}^{u_i(l+1)} = f\left(\frac{e_h^{ik}}{|\mathcal{N}_c|} \sum_{c_j \in \mathcal{N}_c} \mathbf{W}_{c_k}^{u(l+1)} \mathbf{x}^{c_k(l)}\right), \quad (2)
$$

$$
\mathbf{x}^{u_i(l+1)} = \mathbf{MLP}\left(\left[\mathbf{a}_{\mathcal{N}_u}^{u_i(l+1)}; \mathbf{a}_{\mathcal{N}_c}^{u_i(l+1)}; \mathbf{x}^{u_i(l)}\right]\right),
$$

where $\mathbf{x}_{u_j}^{(l)}$ and $\mathbf{x}_{c_k}^{(l)}$ are the user embedding and cascade embedding from the last layer. $\mathbf{W}_{u_j}^{u(l+1)}$ and $\mathbf{W}_{c_k}^{u(l+1)}$ are learnable weight matrices to aggregate contextual features from different neighbors. $\mathcal{N}_u$ and $\mathcal{N}_c$ are set of user node and cascade node neighbors that shares edges with $u_i$. $e_h^{ik}$ and $e_h^{ij}$ is the corresponding edge weights. $f(\cdot)$ means the activation and norm operations.

Similarly, we perform context aggregation operations for cascade nodes. We aggregate the user neighbors to construct cascade embeddings. For the cascade node $c_i$, the process is formulated as,

$$
\mathbf{a}_{\mathcal{N}_u}^{c_i(l+1)} = f\left(\frac{e_h^{ij}}{|\mathcal{N}_u|} \sum_{u_j \in \mathcal{N}_u} \mathbf{W}_{u_j}^{c(l+1)} \mathbf{x}^{u_j(l)}\right),
$$

$$
\mathbf{x}^{c_i(l+1)} = \mathbf{MLP}\left(\left[\mathbf{a}_{\mathcal{N}_u}^{c_i(l+1)}; \mathbf{x}^{c_i(l)}\right]\right), \quad (3)
$$

where $\mathbf{x}_{u_j}^{(l)}$ is the user node embedding from the last layer. $\mathbf{W}_{u_j}^{(l+1)}$ is a learnable weight matrix to aggregate contextual features from user neighbors. $\mathcal{N}_u$ is a set of user nodes that share bipartite relations with cascade node $c_i$.

As our model leverages the same GNN layer at different modules to aggregate structural contexts, we denote the above GNN layer as function $\Psi(\mathcal{G}, \mathbf{X})$, where graph $\mathcal{G}$ and node embeddings $\mathbf{X}$ is the input of GNN layers. Thus, the GNN layer to get node initial state is $\Psi_0^d(\mathcal{G}_h, \mathbf{X}^{d(0)})$, where the input node embedding matrix

$\mathbf{X}^{d(0)} \in \mathbb{R}^{(|\mathcal{U}|+|C|) \times d}$ is generated from a normal distribution [7]. The embedding matrix to represent the initial state of nodes is $\mathbf{X}_0^d$.

After obtaining the initial states for nodes, we characterize the continuous dynamics of the diffusion process with ODE functions. In diffusion processes, the states of users and their relations are deeply correlated and could affect each other. For example, if a user is interested in a piece of information and forwards it to his social friends, the relationship between these users will be closer. Similarly, if a user shares tight relationships with others, he is more likely to be influenced by them and obtain information from them, which will affect his preferences. Therefore, we propose to infer the future state of users and their relations with different but coupled ODE functions.

Typically, the state of edges is determined by the user it connects and their initial attributes. We concatenate the node's initial state to represent the initial attributes of the edges and leverage the current node state to infer edge states. Based on the current state of edges, we could further update the graph structures to obtain future states of users. Thus, the ODE for edges is defined as,

$$
\mathbf{x}_0^{i \to j} = [\mathbf{x}_0^{u_i, d}; \mathbf{x}_0^{u_j, d}],
$$

$$
\frac{d\mathbf{x}_t^{i \to j}}{dt} = \mathbf{MLP}_e\left(\left[\mathbf{x}_i^t \| \mathbf{x}_j^t\right]\right) + \mathbf{MLP}_{\text{init}}\left(\mathbf{x}_0^{i \to j}\right), \quad (4)
$$

$$
e_t^{ij} = \mathbf{MLP}_{\text{weight}}\left(\mathbf{x}_t^{i \to j}\right),
$$

where $\mathbf{x}_0^{i \to j}$ is the initial edge attributes. $[;]$ means concatenation operation. $\mathbf{x}_t^{i \to j}$ represents the edge states at timestamp $t$. $e_t^{ij}$ is the new edge weights at timestamp $t$. The graph structure at timestamp $t$ is $\mathcal{G}_h^t = \left\{\mathcal{U}_h, \mathcal{E}_h, \mathcal{W}_h^t\right\}$.

Intuitively, the state of users is determined by their neighbors and diffusion preference bias at the current timestamp. Moreover, their preference bias at the initial state also deeply affects their preferences in the future. Thus, we define the ODE function for

nodes as follows,

$$\frac{dX_t^d}{dt} = \Psi_d \left( \mathcal{G}_h^t, X_t^d \right) - X_t^d + X_0^d \tag{5}$$

where $X_t^d \in \mathbb{R}^{(|\mathcal{U}|+|C|) \times d}$ denotes the matrix representing all node states at timestamp $t$. $\mathcal{G}_h^t$ is the hetero generous graph with new edge weights at timestamp $t$.

Since we have modeled the dynamics of nodes and edges in the diffusion process with Eq. 4 and Eq. 5. Given a continuous time $t$, the value of nodes and edges can then be solved by a designated ODE solver:

$$X_t^d = \text{ODESolver} \left( \frac{dX_t^d}{dt}, X_0^d, t \right)$$

$$x_t^{i \to j} = \text{ODESolver} \left( \frac{dx_t^{i \to j}}{dt}, x_0^{i \to j}, t \right) \tag{6}$$

To accurately and comprehensively describe users' dynamic preferences, we take $N$ solving steps for the ODE solver and obtain user hidden states $\left\{ X_1^d, X_2^d, \ldots, X_N^d \right\}$ at different timestamps. We develop a channel attention mechanism to integrate users' hidden state to infer users' dynamic preferences. Formally, the users' dynamic preference embedding can be computed as,

$$\alpha^i = \frac{\exp \left( a \cdot W_a X_i^d \right)}{\sum_{j \in N} \exp \left( a \cdot W_a X_j^d \right)}, X^d = \sum_{i \in N} \alpha^i X_i^d, \tag{7}$$

where $a \in \mathbb{R}^d$ and $W_a \in \mathbb{R}^{d \times d}$ are trainable parameters. User dynamic preference embedding matrix is denoted as $X^d \in \mathbb{R}^{|\mathcal{U}| \times d}$.

*4.1.2 Static Correlation Encoding.* In the diffusion process, user behaviors do not strictly abide by their preferences. Instead, they may also follow specific interaction patterns. For example, although social advertisers may show preferences for different products at different times, they tend only to forward information to potential buyers and reject sharing other information. Therefore, we extract static user correlations from a global perspective to complement users' dynamic preferences.

Since heterogeneous graph $\mathcal{G}_h$ contains all historical user interactions and social relations, we directly apply two layers of GNN to extract users' correlation from the heterogeneous graph, which is,

$$X^{s(2)} = \Psi_2^s \left( \mathcal{G}_h, \Psi_1^s \left( \mathcal{G}_h, X^{s(0)} \right) \right), \tag{8}$$

where input node embedding matrix $X^{s(0)} \in \mathbb{R}^{(|\mathcal{U}|+|C|) \times d}$ is randomly initialize with normal distribution [7]. As cascade embeddings hardly provide help for prediction, we only collect the user embedding from $X^{s(2)}$ as static correlation embedding matrix $X^s \in \mathbb{R}^{|\mathcal{U}| \times d}$.

We introduce a gated fusion strategy to integrate users' dynamic preferences with static correlations. For user $u_j$, his embeddings $x_{u_j}$ is derived from the following procedure,

$$g_j = \sigma \left( x^{u_j, d} W_d + x^{u_j, s} W_s \right),$$

$$x^{u_j} = g_j x^{u_j, d} + (1 - g_j) x^{u_j, s}, \tag{9}$$

where $W_s$ and $W_d$ are trainable parameters. We perform the same operation for each user and obtain the user embedding matrix $X^u$.

## 4.2 Cascade Representation

The future diffusion path of a cascade is affected by its diffusion pattern, which can be reflected through the order of previously infected users and infected timestamps in the cascade, *i.e.*, user context and temporal context. In this section, we represent the diffusion pattern in the observed cascade according to its user and temporal contexts to assist prediction.

*4.2.1 Temporal Context Representation.* Empirical studies [2, 10] have shown that the influence of diffusion cascades decreases over time, which is known as the time-decay effect. They consider the diffusion process as a temporal point process and estimate the activation timestamp through the time difference between each user interaction. Inspired by them, we propose to capture the temporal context by mapping the time difference into vector space with a neural function.

For timestamp sequence $c_o^t = \{t_1^o, t_2^o, ..., t_{L_o}^o\}$ of observed cascade $c_o$, we characterize its temporal context by the time difference between each diffusion behavior, which is represented as $\Delta t_j^o = t_{i+1} - t_i, 1 < i < L_{c_o}$. We map each time difference $\Delta t_i^o$ into a new time label based on the constructed time intervals. The time label is calculated by following the function,

$$\lambda_i^o = \left\lfloor \frac{\Delta t_i^o - t_{min}}{\lceil (t_{max} - t_{min}) / l_t \rceil} \right\rfloor, \tag{10}$$

where $t_{max}, t_{min}$ are predefined maximum and minimum time differences. $l_t$ is the number of time slots used to discretize the time difference in the observed cascade. We transform $\lambda_i^o$ into one-hot embedding $t_i^o \in \mathbb{R}^{l_t}$. Each element in $t_i^o$ is set to 0, except for the element in position $\lambda_i^o$ is set to 1. Finally, we generate temporal context encoding by capturing the time-decay pattern via an MLP layer,

$$z_i^{o,t} = \tanh \left( W_t t_i^o + b_t \right), \tag{11}$$

where $W_t \in \mathbb{R}^{l_t \times d}$ and $b_t \in \mathbb{R}^d$ are learnable parameters. $\tanh(\cdot)$ is the activation function.

*4.2.2 User Context Representation.* Although we have encoded users into embeddings, these embeddings contain massive contextual information from different relations and structures, which hardly reflects the specific diffusion pattern in the observed cascade. Thus, we incorporate a heuristic self-attention mechanism to filter out relevant information among participants and aggregate them to represent specific user contexts.

For the user sequence $c_o^u = \{u_1^o, u_2^o, ..., u_{L_o}^o\}$ of the observed cascade, we first look up the user embedding matrix $X^u$ to transform it into the user embedding sequence $z_o^u = \left[ x_1^o, x_2^o, \ldots, x_{L_{c_o}}^o \right]$. Then, we compute relevant scores between each user $u_j \in c_o^u$ and his context user $u_k \in \{u_1, ..., u_{j-1}\}$ and apply the weighted attention score sum to aggregate the relevant information. Specifically, the context-enhanced embedding $\hat{x}_j^o$ for $u_j$, is calculated as,

$$\beta_{kj} = \frac{\exp \left( \sigma(W_k x_k^o) \odot \sigma(W_j x_j^o) \right)}{\sum_{r=1}^{j-1} \exp \left( \sigma(W_r x_r^o) \odot \sigma(W_j x_j^o) \right)},$$

$$\hat{x}_j^o = \sum_{k=1}^{j-1} \beta_{kj} x_j^o, \tag{12}$$

where $\beta_{kj}$ denotes the relevance score between user $u_j$ and $u_k$. $\mathbf{W}_k^d, \mathbf{W}_j^d \in \mathbb{R}^{d \times d}$ are transformation matrices to map the user embeddings into the different linear spaces to measure their correlation. $\sigma(\cdot)$ is the activation function. $\odot$ denotes the Hadamard product operation for user embeddings.

Moreover, information diffusion is a stochastic process driven by multiple factors [36], which does not always abide by the diffusion patterns learned from previously infected users. To comprehensively characterize the user context in the observed cascade, we apply a recursive residual connection layer to fuse the context-enhanced embedding sequence with the user embedding sequence to extend the contextual information for each user. Specifically, for user $u_j$, the user context representation is derived from the following procedure,

$$\mathbf{z}_j^{o,u} = \mathbf{LN}(\mathbf{x}_j^o + \mathbf{LN}(\mathbf{x}_j^o + \hat{\mathbf{x}}_j^o)) \tag{13}$$

where $\mathbf{LN}(\cdot)$ means the layer normalization.

Finally, we concatenate temporal and user contexts to represent the observed cascade. For user $u_j$, his representation $\mathbf{z}_j^o$ is computed as,

$$\mathbf{z}_j^o = [\mathbf{z}_j^{o,u}; \mathbf{z}_j^{o,t}], \tag{14}$$

where $[ ; ]$ is the concatenate operation. The observed cascade $c_o$ is represented as $\mathbf{Z}^o = \left[ \mathbf{z}_1^o, \mathbf{z}_2^o, \ldots, \mathbf{z}_{L_{c_o}}^o \right] \in \mathbb{R}^{L_{c_o} \times d}$.

## 4.3 Prediction

Although we have represented the observed cascade as $\mathbf{Z}^o$ by jointly exploring its user and temporal contexts, the context-dependence between each user is still unclear to achieve the prediction. Therefore, we apply a multi-head decoding layer to attend to different contextual information in the observed cascade efficiently. The process could be formulated as follows:

$$\mathbf{Attention}(\mathbf{Q}, \mathbf{K}, \mathbf{V}) = \mathbf{softmax}\left( \frac{\mathbf{Q}\mathbf{K}^T}{\sqrt{d_h}} + \mathbf{M} \right)\mathbf{V},$$
$$\mathbf{o}_i^d = \mathbf{Attention}\left( \mathbf{Z}^o\mathbf{W}_i^Q, \mathbf{Z}^o\mathbf{W}_i^K, \mathbf{Z}^o\mathbf{W}_i^V \right), \tag{15}$$
$$\mathbf{Z}^h = [\mathbf{o}_1^d; \mathbf{o}_2^d; \ldots; \mathbf{o}_H^d]\mathbf{W}^O,$$

where $\mathbf{W}_i^Q, \mathbf{W}_i^K, \mathbf{W}_i^V \in \mathbb{R}^{d \times d_h}$, and $\mathbf{W}^O \in \mathbb{R}^{H \times d_h \times d}$ are learnable parameters. $H$ is the number of heads in the multi-head self-attention module. $d_h$ is the scaling factor. $\mathbf{M} \in \mathbb{R}^{L_{c_o} \times L_{c_o}}$ is a matrix to mask out future users to avoid label leakage, which is denoted as,

$$\mathbf{M}_{ij} = \begin{cases} 0 & \text{otherwise,} \\ -\infty & i \geq j. \end{cases} \tag{16}$$

Then, we apply two layers of fully connected neural networks to obtain attentive cascade representation $\mathbf{Z}^p$:

$$\mathbf{Z}^p = \sigma(\mathbf{Z}^h\mathbf{W}_1^h + \mathbf{b}_1)\mathbf{W}_1^h + \mathbf{b}_2 \tag{17}$$

where $\mathbf{W}_1^h, \mathbf{W}_2^h$ are all learnable transformation matrices. $\mathbf{b}_1, \mathbf{b}_2$ are bias parameters.

Finally, we use the predicted cascade representations $\mathbf{Z}_p$ to calculate infected probabilities $\hat{\mathbf{y}}_{ij} \in \mathbb{R}^{L_{c_o} \times |\mathcal{U}|}$ for all users, i.e.,

$$\hat{\mathbf{y}}_{ij} = \mathbf{softmax}(\mathbf{W}^p\mathbf{Z}^p + \mathbf{Mask}) \tag{18}$$

where $\mathbf{W}_p \in \mathbb{R}^{d \times d}$ is a learnable parameter to calculate the infect probability for each user. We utilize $\mathbf{Mask} \in \mathbb{R}^{|\mathcal{U}| \times L_{c_o}}$ matrix to mask users who have already been activated in the observed cascade sequence. We adopt cross-entropy loss as the objective to optimize the information diffusion prediction task:

$$\mathcal{L}(\theta) = - \sum_{i=2}^{|c_o|} \sum_{j=1}^{|\mathcal{U}|} \mathbf{y}_{ij} \log(\hat{\mathbf{y}}_{ij}), \tag{19}$$

where $\theta$ represents all parameters that need to be learned in the model $\mathbf{y}_{ij} = 1$ denotes that the predicted user $u_j$ is infected at timestamp $t_i^o$, otherwise $\mathbf{y}_{ij} = 0$.

# 5 EXPERIMENT

## 5.1 Experimental Setups

*5.1.1 Datasets.* We incorporate four publicly available real-world datasets to evaluate the performance of our model. The detailed statistics of the datasets are presented in Table 2. (1) **Twitter** [9] records the tweets with URLs during October 2010 on Twitter and its diffusion paths. The social relations are pre-defined by the following relation on Twitter. (2) **Douban** [19] is collected from a Chinese social website named Douban, where people can share their book reading statuses. The co-occurrence connection of users is interpreted as their social relations. (3) **Android** [18] is collected from Stack-Exchanges, a community Q&A website. Cascade refers to a series of chronologically ordered posts associated with the tag "Android". The social relation is pre-defined by user interactions, *e.g.*, answering or commenting on the same post. (4) **Meme-tracker**[1] [12] tracks the migration of frequent quotes and phrases, *i.e.* memes. Each URL is treated as a user in the dataset.

*5.1.2 Baselines.* To evaluate the performance of GODEN, we select eight information diffusion prediction models as baselines for comparison. We preserve the original parameter settings for each model. The baseline models are (1) **NDM** [32] utilizes the self-attention mechanism and convolution modules to attend to long-term user correlation in cascade sequences. (3) **FOREST** [33] is a recurrent model that employs GRU to learn sequential features and extracts network structure information via GCN. (4) **CEGCN** [22] utilizes GNNs to exploit collaborative patterns from other cascades for prediction. (5) **DyHGCN** [35] applies GCN to learn users' dynamic preferences by discretising the diffusion process into heterogeneous subgraphs (6) **MS-HGAT** [19] constructs a series of hyper-graphs to model user interactions and integrate them with static social relations to depict interaction dependencies among users. (7) **DisenIDP** [5] leverages two hyper GCN to learn users' intents in the diffusion process and designs a self-supervised disentanglement task to assist the procedure. (8) **RotDiff** [15] maps the users into the hyperbolic representation space based on the social relations and diffusion paths, which achieves state-of-the-art performances.

*5.1.3 Implementation Details.* We implement our model in PyTorch and conduct our experiments on an Ubuntu server equipped with two 32 GB Nvidia V100 GPUs. GODEN is trained based on the Adam optimizer with parameters $\beta_1$ and $\beta_2$ set to 0.90 and 0.99, respectively. The learning rate is set as 0.001. The batch size in

---

[1]http://memetracker.org/

**Table 1: Experimental results on HITS score over four datasets (%).**

| Models | Android | | | Memetracker | | | Twitter | | | Douban | | |
|---|---|---|---|---|---|---|---|---|---|---|---|---|
| | H@10 | H@50 | H@100 | H@10 | H@50 | H@100 | H@10 | H@50 | H@100 | H@10 | H@50 | H@100 |
| NDM | 0.0339 | 0.0953 | 0.1572 | 0.2083 | 0.3663 | 0.4583 | 0.1934 | 0.2941 | 0.3573 | 0.1013 | 0.2123 | 0.3125 |
| FOREST | 0.0700 | 0.1514 | 0.2237 | 0.2963 | 0.4780 | 0.5786 | 0.2552 | 0.3850 | 0.4607 | 0.1868 | 0.3084 | 0.3857 |
| CEGCN | 0.1075 | 0.2109 | 0.2842 | 0.2951 | 0.5021 | 0.6112 | 0.3381 | 0.5140 | 0.5987 | 0.2078 | 0.3483 | 0.4267 |
| DyHGCN | 0.0842 | 0.1915 | 0.2679 | 0.2952 | 0.4864 | 0.5848 | 0.2901 | 0.4688 | 0.5719 | 0.1987 | 0.3289 | 0.3942 |
| MS-HGAT | 0.1049 | 0.1987 | 0.2781 | 0.2843 | 0.4966 | 0.6047 | 0.2996 | 0.4654 | 0.5735 | 0.2065 | 0.3504 | 0.4136 |
| DisenIDP | 0.0946 | 0.1916 | 0.2684 | 0.3074 | 0.5199 | 0.6280 | 0.3273 | 0.4799 | 0.5540 | 0.2059 | 0.3545 | 0.4284 |
| RotDiff | 0.1144 | 0.2304 | 0.3130 | 0.3066 | 0.5170 | 0.6206 | 0.3590 | 0.5246 | 0.6121 | 0.2216 | 0.3823 | 0.4637 |
| GODEN | **0.1201** | **0.2401** | **0.3269** | **0.3379** | **0.5430** | **0.6399** | **0.3811** | **0.5578** | **0.6475** | **0.2490** | **0.3926** | **0.4729** |
| Improve.(%) | 4.98 | 4.21 | 4.44 | 10.21 | 5.03 | 3.11 | 6.16 | 6.33 | 5.78 | 12.36 | 2.69 | 1.98 |

**Table 2: Statistics of the datasets.**

| Datasets | Twitter | Douban | Android | Memetracker |
|---|---|---|---|---|
| # Users | 12,627 | 12,232 | 9,958 | 4,709 |
| # Social Links | 309,631 | 348,280 | 48,573 | - |
| # Cascades | 3,442 | 3,475 | 679 | 12,661 |
| Avg. Repost | 10.74 | 6.18 | 2.27 | 44.00 |
| Avg. Length | 32.60 | 21.76 | 33.3 | 16.24 |

the training set is 16. The dimension of user embeddings is all set to $d = 64$. For user dynamic preference embeddings, we set the hidden dimensions of GNN $\Psi_0^d(\cdot)$ as 128. The hidden state of edge and nodes in ODEs are 256 and 128, respectively. We use Runge-Kutta-4(RK-4) as the solver of our coupled ODE. For user static correlation embeddings, we set the hidden dimensions of GNN $\Psi_1^s(\cdot)$ and $\Psi_1^s(\cdot)$ as 128 and 64, respectively. The time interval $l_t$ to map the time difference in the observed cascades into vector space is set to 5000. The dimensionality of temporal context encoding $d_t$ is set to 8. The number of heads $H$ in a multi-head attention module is chosen from $\{8, 10, 12, 16\}$ and set to 8 after comparison. For all four datasets, we randomly sample 80% of cascades for training and split the remaining 20% evenly for validation and testing. The maximum cascade length is set to 200 for all datasets. Since information diffusion prediction aims to predict user participation by ranking all uninfected users according to their infection probabilities, Following the evaluation protocol of previous works [19, 35], we consider the task as an information retrieval task and evaluate the performance of information diffusion prediction models with two ranking metrics, *i.e.*, Mean Average Precision on top $K$ and HITS scores on top $K$, with $K = [10, 50, 100]$. We abbreviate them as MAP@K (M@K) and Hits@K (H@K), respectively.

## 5.2 Experimental Results

The experimental results of the information diffusion prediction task are shown in Table 1 and Table 3. Numbers in bold denote the best results among all models and the underlined ones denote the second best results. Improvements in GODEN are statistically significant with $p < 0.01$ on paired $t$-test. With the result, we have the following observations, **(O1)** GODEN consistently and significantly outperforms all state-of-the-art baselines on all four datasets under

different evaluation metrics. The relative improvements over the best-performing baseline are at least 6.18% **(O2)** Generally, methods applying graph data to explore user correlations beyond cascade sequence perform well. Instead, NDM focuses on learning user correlation in the sequence with attention mechanism gets limited performance. FOREST, CEGCN, and DisenIDP extend user relations with graph structure to improve prediction performance. DyHGCN and MS-HGAT achieve relatively high performance by creating a series of graph snapshots to describe the diffusion process. RotDiff represents users in hyperbolic space to predict infection probability, which achieves better performance than most methods based on Euclidean space. **(O3)** GODEN achieves significant improvements compared to all baselines. We attribute the improvement to two reasons. For one, we leverage coupled ODE to model the continuous dynamics of users and relations in the diffusion process, which allows GODEN to infer users' preferences accurately. For another, we represent the diffusion pattern based on user and temporal context in the observed cascade. This specific context information could assist GODEN in retrieving users with similar preferences and filtering out irrelevant users.

## 5.3 Ablation Study Results

To validate the contribution of each component in GODEN, we design six variants for our model, which are 1) GODEN$_{-DP}$ removes user dynamic preference embedding, which only utilizes the user static correlation embedding for prediction. 2) GODEN$_{-SC}$ removes the static correlation embedding of the user, which only utilizes the dynamic preference embedding of the user for prediction. 3) GODEN$_{-CA}$ removes channel attention to fuse multiple user preferences embeddings and only solves coupled ODEs in one step. 4) GODEN$_{-ODE}$ removes the ODE solver and leverages 3 layers of GNN and MLP layer to model the dynamics of nodes and edges. 5) GODEN$_{-EdgeODE}$ removes edge ODE function and solves Eq. 5 with fixed graph structure. 6) GODEN$_{-UC}$ removes the user context embedding in the cascade representation module and directly fuses user embedding and temporal context for prediction.

The results of these variants are shown in Table 4. By analyzing the results, we have the following observations: (1) All variants suffer performance drops compared with GODEN, which shows that each component is essential for prediction. Moreover, GODEN$_{-UC}$

**Table 3: Experimental results on MAP score over four datasets.**

| Models | Android | | | Memetracker | | | Twitter | | | Douban | | |
|---|---|---|---|---|---|---|---|---|---|---|---|---|
| | M@10 | M@50 | M@100 | M@10 | M@50 | M@100 | M@10 | M@50 | M@100 | M@10 | M@50 | M@100 |
| NDM | 0.0160 | 0.0202 | 0.0210 | 0.0931 | 0.1031 | 0.1048 | 0.1169 | 0.1243 | 0.1256 | 0.0581 | 0.0651 | 0.0663 |
| FOREST | 0.0381 | 0.0416 | 0.0426 | 0.1553 | 0.1637 | 0.1751 | 0.1733 | 0.1790 | 0.1801 | 0.1086 | 0.1146 | 0.1183 |
| CEGCN | 0.0706 | 0.0750 | 0.0760 | 0.1654 | 0.1749 | 0.1765 | 0.2107 | 0.2189 | 0.2202 | 0.1202 | 0.1267 | 0.1278 |
| DyHGCN | 0.0633 | 0.0675 | 0.0685 | 0.1542 | 0.1641 | 0.1657 | 0.1880 | 0.1951 | 0.1965 | 0.1122 | 0.1187 | 0.1198 |
| MS-HGAT | 0.0458 | 0.0503 | 0.0514 | 0.1611 | 0.1623 | 0.1725 | 0.1751 | 0.1832 | 0.1847 | 0.1048 | 0.1114 | 0.1148 |
| DisenIDP | 0.0582 | 0.0623 | 0.0634 | 0.1624 | 0.1722 | 0.1737 | 0.2159 | 0.2228 | 0.2239 | 0.1041 | 0.1110 | 0.1121 |
| RotDiff | 0.0696 | 0.0745 | 0.0756 | 0.1653 | 0.1691 | 0.1766 | 0.2406 | 0.2482 | 0.2495 | 0.1170 | 0.1254 | 0.1266 |
| GODEN | **0.0739** | **0.0792** | **0.0804** | **0.1939** | **0.2036** | **0.2050** | **0.2480** | **0.2562** | **0.2575** | **0.1486** | **0.1553** | **0.1564** |
| Improve.(%) | 6.18 | 6.31 | 6.35 | 17.3 | 20.4 | 16.08 | 3.08 | 3.22 | 3.21 | 27.01 | 23.84 | 23.54 |

**Table 4: Ablation study on three datasets.**

| Models | Twitter | | | | Douban | | | | Android | | | |
|---|---|---|---|---|---|---|---|---|---|---|---|---|
| | H@50 | H@100 | M@50 | M@100 | H@50 | H@100 | M@50 | M@100 | H@50 | H@100 | M@50 | M@100 |
| GODEN | **0.5578** | **0.6475** | **0.2480** | **0.2562** | **0.3926** | **0.4729** | 0.1553 | 0.1564 | **0.2401** | **0.3269** | **0.0739** | **0.0792** |
| GODEN$_{DP}$ | 0.5328 | 0.6275 | 0.2327 | 0.2340 | 0.3822 | 0.4575 | 0.1467 | 0.1478 | 0.2042 | 0.2873 | 0.0708 | 0.0719 |
| GODEN$_{SC}$ | 0.5161 | 0.6202 | 0.2209 | 0.2224 | 0.3545 | 0.4340 | 0.1393 | 0.1404 | 0.1957 | 0.2911 | 0.0716 | 0.0730 |
| GODEN$_{CA}$ | 0.5254 | 0.6403 | 0.1822 | 0.1839 | 0.3921 | 0.4712 | **0.1600** | **0.1612** | 0.2352 | 0.3137 | 0.0752 | 0.0764 |
| GODEN$_{ODE}$ | 0.5176 | 0.6280 | 0.2001 | 0.2016 | 0.3434 | 0.4206 | 0.1211 | 0.1222 | 0.2089 | 0.2849 | 0.0689 | 0.070 |
| GODEN$_{EdgeODE}$ | 0.5305 | 0.6244 | 0.2480 | 0.2494 | 0.3755 | 0.4527 | 0.1480 | 0.1491 | 0.2112 | 0.2888 | 0.0728 | 0.0749 |
| GODEN$_{UC}$ | 0.5366 | 0.6139 | 0.2405 | 0.2506 | 0.3440 | 0.4234 | 0.1195 | 0.1206 | 0.2065 | 0.2888 | 0.0703 | 0.0715 |

suffers a significant performance drop in HIT scores, which suggests that it is important to consider specific diffusion patterns in the cascade to retrieve potential users. (2) Compared with GODEN, the performance of variants GODEN$_{DP}$ and GODEN$_{SC}$ shows that removing any type of user encoding mechanism would lead to performance degradation. These results indicate that users' dynamic preferences and static correlations are both key factors affecting the diffusion process. (3) When we remove the ODE function on edges(GODEN$_{EdgeODE}$), the model suffers from certain drops. This result verifies that the hidden dynamic of users in the diffusion process is influenced by their relations. When we remove the ODE solver and estimate diffusion dynamics with only neural networks(GODEN$_{ODE}$), the model shows worse performance, highlighting the importance of estimating hidden dynamics with ODE functions instead of discrete neural networks.

## 5.4 Parameter Analyze Results

In this subsection, we conduct comparative experiments on the Douban and Android datasets and further analyze the effect of maximum cascade length. The result in Figure 3 shows that our model could outperform other models in any cascade length, illustrating its stability and effectiveness. We contribute its remarkable performance to our coupled ODE module, which could model the hidden dynamics of the diffusion process and infer users' dynamic preferences accurately regardless of cascade length and duration.

## 6 CONCLUSION

In this work, we propose a novel graph ordinary differential equation network (GODEN) for information diffusion prediction, which

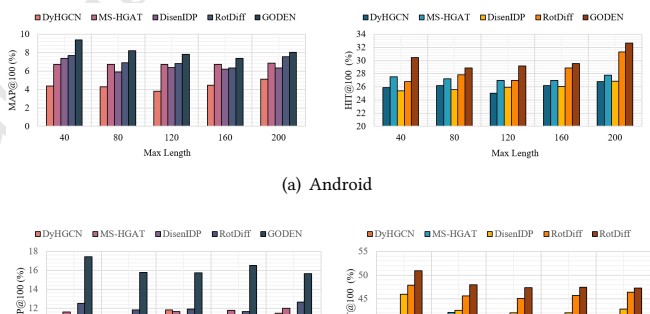

(a) Android

(b) Douban

**Figure 3: Impact of maximum cascade length.**

models the continuous dynamics of the diffusion process. With the coupled ODEs to characterize the co-evolution dynamics of users and their relations, GODEN could accurately infer users' dynamic preferences. Moreover, we extract static user correlation from the heterogeneous graph to complement users' dynamic preferences. To predict the future infection probability, we first represent the diffusion pattern in the observed cascade based on its temporal and user contexts. Then, we leverage a multi-head attention mechanism to attend to different contexts. The experimental results demonstrate the effectiveness of GODEN.

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
