# OpenReview forum: "Information Diffusion Prediction with Graph Neural Ordinary Differential Equation Network"
_acmmm.org/ACMMM/2024/Conference — MM2024 Poster_

### Official Review · Reviewer_ji9d · 2024-05-20

**Rating:** 4
**Confidence:** 3

**Summary:**

The paper is about information diffusion prediction, which is an interesting topic. The authors propose a novel Graph Neural Ordinary Differential Equation Network for information diffusion prediction, which incorporates neural ordinary differential equations (ODE) to model the continuous dynamics of the diffusion process. The paper is well written and well organized. However, there are several concerns in the current version of the paper that addressing them will increase the quality of this paper.

**Strengths:**

[1] Novel ideas and research questions. [2] Reasonable writing logic. [3] sufficient experimental results.

**Limitations:**

1. My question is, although the data sets used in the article do come from different fields (or modalities), are there modalities other than graph structures on the actual training data?

2. I hope to see a discussion on computational complexity. At present, it seems that the design of the model is effective, but there seem to be more superimposed modules. Will these different modules have an impact on the core complexity of the model?

3. The improvement of the model on different data sets varies greatly. I hope the author can further explain this phenomenon.

4. Some related work of graph learning on ACM MM can be considered.
[1] Pre-training graph transformer with multimodal side information for recommendation. ACM MM 2021.
[2] Multi-modal graph contrastive learning for micro-video recommendation. ACM MM 2022.
[3] TMac: Temporal multi-modal graph learning for acoustic event classification. ACM MM 2023.

**Suitability:**

1

---

### Official Review · Reviewer_HEUv · 2024-05-21

**Rating:** 2
**Confidence:** 4

**Summary:**

This paper proposes a novel approach to predict the spread of information in social networks. Unlike previous methods that discretize the diffusion process or summarize user preferences based on partially observed snapshots, GODEN employs neural ordinary differential equations (ODEs) to model the continuous dynamics of user preferences and relationships. By integrating user correlations from a heterogeneous graph and employing a multi-head attention mechanism, GODEN aims to enhance the accuracy of predicting future participants in information cascades. Experimental results on four real-world datasets demonstrate that GODEN significantly outperforms existing state-of-the-art models.

**Strengths:**

1.	The authors propose a novel Graph Neural Ordinary Differential Equation Network(Goden) for information diffusion prediction, which incorporates neural ordinary differential equations(ODE) to model the continuous dynamics of the social information diffusion process.

2.	In this paper, the static correlation of users (that is, user contact in the network structure) is extracted to extend the dynamic preferences of users over time, and the specific diffusion pattern of the cascade is further represented by learning the user context and time context information.

3.	The authors combine ODE with a graph neural network, give a full concept and formal description, and make a relatively rich experimental evaluation.

**Limitations:**

1. The authors claimed that the content in a social network involves multimodal information such as text, images, video, etc., they need to be considered together when making predictions in the social network. However, in this paper, the authors did not take into account the multimedia content of information on social networks and did not present models or methodologies tailed for multimedia information prediction. In the experiment section, the datasets also do not contain multimedia content. It is thus not suitable for the MM conference.

2. The use of ODE modeling to predict information transmission originates from the infectious disease model. As we all know, ODE modeling has the advantage of good prediction effect and high computational efficiency (focusing on time cost). The ODE and GNN, Attention, MLP, and other classical neural network architecture combined modeling significantly increase model complexity.

3. In Section 4.1 User Encoding, the model uses traditional GNN for message passing between user nodes. Why not use other GNN encoders such as GCN and GAT? Compared with traditional GNN, they consider the degree of nodes or directly introduce the attention mechanism, and the theoretical effect should be better. Is the traditional GNN adopted for the sake of task or data fitting? The experiment could be enhanced by including a comparison of different GNNs.

4. In section 5.4 Parameter Analyze Result, the title of Parameter Analysis will be better. Furthermore, the scope of the parameter sensitivity analysis experiment is insufficient, focusing solely on examining the impact of varying maximum cascade selection lengths on experimental outcomes. For instance, within Section 4.2.1 titled "Temporal Context Representation," formula (10) introduces the %l_t% parameter, delineating its role in time encoding through the selection of different time slots. Similarly, it is imperative to scrutinize a broader range of core parameters associated with the model and thoroughly discuss their respective influences on model performance.

**Suitability:**

1

---

### Official Review · Reviewer_CnVC · 2024-05-25

**Rating:** 4
**Confidence:** 3

**Summary:**

This paper proposes a novel Graph Neural Ordinary Differential Equation Network (GODEN) for information diffusion prediction, which incorporates neural ordinary differential equations (ODE) to model the continuous dynamics of the diffusion process.

**Strengths:**

1. This paper proposes two coupled ODEs to learn the evolution pattern for users and relations in the diffusion process and infer users’ dynamic preferences.
2. This paper extracts users’ static correlations to extend their dynamic preferences and represents the specific diffusion pattern of cascades by learning the user context and temporal context information to promote the prediction.

**Limitations:**

1. The method mentioned in this paper does not involve multimodal. The matching degree with the meeting is low.
2. From the overall structure of the method, we can see that the main innovation is to apply ordinary differential equations to information distribution prediction, but from the results of ablation experiments, we can see that the performance of GODEN-ODE and GODEN-EdgeODE is not the worst. Does it reflect that the benefits brought by ODE are limited?
3. The structure of the heterogeneous graph is unclear, and the benefits of constructing the heterogeneous graph are not reflected in the ablation experiment.
4. This paper does not provide source code to verify the feasibility of the method used in the paper.

**Suitability:**

2

---

### Meta-Review · Area_Chair_otz7 · 2024-07-05

**Recommendation:** Accept (Poster)
**Confidence:** 4

**Metareview:**

The paper proposes a novel approach, GODEN, to predict information diffusion in social networks using neural ordinary differential equations. Still, it lacks consideration of multimodal content and has high model complexity, making it a suitable for Poster Paper for the MM conference.